# Acute Tongue Swelling as a Still Unexpected Manifestation of Internal Carotid Artery Dissection: A Case Report

**DOI:** 10.3390/brainsci13040603

**Published:** 2023-04-02

**Authors:** Wioletta Pawlukowska, Krystian Mross, Marta Jankowska, Łukasz Zwarzany, Wojciech Poncyljusz, Marta Masztalewicz

**Affiliations:** 1Department of Neurology, Pomeranian Medical University in Szczecin, Unii Lubelskiej 1, 71-252 Szczecin, Poland; 2Department of Diagnostic Imaging and Interventional Radiology, Pomeranian Medical University in Szczecin, Unii Lubelskiej 1, 71-252 Szczecin, Poland

**Keywords:** carotid artery dissection, local sign, ICAD, tongue swelling, neuroimaging, case report

## Abstract

The diagnosis of internal carotid artery dissection (ICAD) at the stage of local signs is essential in the prevention of the life-treating cerebral complication; however, making this diagnosis has significant difficulties. We present the case of a 36-year-old female with left ICAD with asymmetric left-sided tongue swelling as an unusual and still unexpected symptom. The patient’s complaints at admission were left-sided numbness of the tongue and swallowing difficulties but its movements were intact. Despite the provided treatment for suspected angioedema, no improvement was noted. Additional examination revealed left-sided tongue weakness, ipsilateral soft palate palsy, paralysis and reduced tension of the left vocal fold, and left-sided Horner’s syndrome. Another suspected diagnosis was a dysfunction of the IX, X, and XII cranial nerves. A head MRI revealed an intramural hematoma of the left internal carotid artery. The radiologists suggested ICAD. The angio-MRI of the head arteries confirmed this diagnosis. The patient received dual antiplatelet therapy. The neuro-logopaedic therapy was also implemented. Currently, the patient’s symptoms are gradually improving with significantly better results on follow-up neuroimaging. Among the possible local symptoms of ICAD, proper attention should be paid to asymmetric swelling of the tongue as an atypical manifestation of damage to the hypoglossal nerve.

## 1. Introduction

Internal carotid artery dissection (ICAD) is a rare condition with 2.5–3 cases per 100,000 population. However, it carries a high (30–80%) risk of ischaemic neurological complications (e.g., amaurosis fugax, transient ischaemic attack, ischaemic stroke) [1,2]. The condition is also related to the risk of subarachnoid haemorrhage [3].

Ischemic (or, much less frequently, haemorrhagic) neurological episodes may be the first manifestation of the disease but they are usually preceded by local symptoms. These symptoms might appear a few hours, a few days, or sometimes even a few weeks earlier [2,3,4]. The diagnosis of the disease at this stage creates significant difficulties in everyday clinical practice but it is of undeniable importance in the prevention of disabling and life-threatening cerebral complications.

We present the case of a female patient with left internal carotid artery dissection whose local symptoms were difficult to diagnose which delayed the final diagnosis. The patient reached out to our hospital with symptoms suggestive of asymmetric left-sided swelling of the tongue, which retrospectively was a sign of hypoglossal nerve injury but upon initial assessment was interpreted as Quincke’s oedema. She had a unilateral headache the previous day. We present this case to show that recognition of the hypoglossal nerve injury at this stage with a headache on the previous day as an initial symptom, would have likely allowed us to avoid the delay in proper diagnosis in our patient.

The presented case is another example of the possible, but unusual and still unexpected, local symptom of internal carotid dissection with the atypical manifestation of damage to the hypoglossal nerve in the form of asymmetric swelling of the tongue [5,6,7,8,9,10].

## 2. Case Presentation

### 2.1. Patient

The patient presented in this case study was a 36-year-old female without any comorbidities. She was not taking any medications on a regular basis and denied the use of stimulants. The patient had no history of recent infections or head or neck traumas. Nevertheless, the day before the admission, the patient sought neurological care in our hospital due to a left-sided headache accompanied by emesis. At that time, no deviations in the neurological examination were found. Computed Tomography (CT) of the head revealed no pathologies. After receiving symptomatic treatment with improvement, she was discharged.

The following day the patient presented to the Emergency Department with complaints of numbness of the tongue on the left side and difficulties in swallowing that appeared early in the morning. However, she did not report a headache as she had the day before. At the time of her admission, she was in good general condition, and was circulatory and respiratory efficient. Neurological examination revealed dysarthric speech with hoarseness. Additionally, the tongue was asymmetrical but did not deviate when protruded, and its movements were correct. Moreover, no other abnormalities were found. A CT of the head as well as lumbar puncture and cerebrospinal fluid examination revealed no abnormalities.

### 2.2. Diagnosis

The angioedema was taken into account. Hence, the administered treatment included corticosteroids and antihistamine drugs. After being observed for several hours in the ED, her condition did not improve. The continuation of treatment was recommended along with fibreoptic examination.

On further examination after another few hours, the tongue movements were impaired with limited mobility to the left side. The soft palate deviated to the right during phonation. The fibreoptic examination disclosed left vocal fold paralysis with medial positioning as well as reduced tension of the left vocal fold. The symptoms corresponded to the damage to IX, X, and XII cranial nerves. Moreover, Horner’s syndrome appeared on the left side.

The diagnostic process was extended by Magnetic Resonance Imaging (MRI) of the head with contrast agent administration. It presented no abnormalities in the brain but allowed the identification of a crescent-shaped thickening of the left internal carotid artery (ICA) wall in the distal cervical segment (C1 according to Bouthillier’s classification). The image corresponding to the dissection with intramural hematoma is shown in Figure 1. The intramural hematoma measured 5 mm in thickness and caused a significant narrowing of the arterial lumen.

Therefore, after being admitted to the Department of Neurology, the patient received conservative treatment including double antiplatelet therapy (clopidogrel 75 mg and acetylsalicylic acid 75 mg as tablets).

Additional magnetic resonance angiography (MRA) of the head arteries confirmed the dissection of the left ICA in the C1-C2 segment with visible intramural hematoma, causing a significant narrowing of the lumen canal from maximum to 90% at the level of the entrance to the carotid canal. It also revealed a 6 × 6 × 2.5 mm pseudoaneurysm just before the entrance of the canal. In the right vertebral artery in the V2 segment on a section of approximately 20 mm in length, a crescent-shaped, hyperintense fringe on the perimeter in T1-weighted images was found, sizing up to 1.5 mm. The image that could correspond to a small intramural hematoma (without accompanying significant narrowing of the lumen canal) is shown in Figure 2. The description of the MRA drew attention to the asymmetry of the tongue. Its base on the left filled the epiglottic vallecula. The left half of the tongue became more intense on post-contrast enhancement (Figure 3). The image could be the result of pressure or damage to the left hypoglossal nerve.

Eventually, the cause of the observed disorders and the final diagnosis was left internal carotid artery dissection. Clinically, the picture was similar to Villaret’s syndrome [11,12]. Retrospectively, it was clear that the left-sided headache experienced by the patient the day before the tongue numbness appeared was the first symptom of the diagnosed vascular pathology.

### 2.3. Treatment

Dual antiplatelet therapy was maintained in treatment. As a response to the increased blood pressure values, hypotensive treatment was also introduced (ramiprilum 5 mg as a tablet). The risk of irritation of the aneurysm during stimulation made it impossible to implement neuro-logopaedic therapy at this point. The patient received instructions on how to improve swallowing.

In the following days of the stay, the Horner’s syndrome subsided. Moreover, the soft palate palsy and tongue weakness decreased. There was also a slight improvement in swallowing but the speech was voiceless and hoarse with great breathing effort and fatigue.

The patient stayed in the Neurology Clinic for seven days. Recommendations on discharge were to maintain double antiplatelet therapy and to schedule the follow-up MRA of the cerebral arteries in three months.

### 2.4. Possible Background and Risk Factors

Basing on additional laboratory tests, we found no data for the infectious or post-infectious background of the diagnosed vascular pathology. The results of the examination for coagulopathy were normal. No data were found for connective tissue disease. The CT-angiography (CTA) of the aorta and renal arteries did not reveal any evidence of fibromuscular dysplasia. No data were found for this disease in the image of the cerebral arteries [13,14,15,16]. In addition, there were no radiological features of Eagle syndrome [17] based on the CTA of the neck (Figure 4). The patient denied any direct relationship between the occurrence of her ailments and physical exertion. It is worth noting that she had been training for several years at the gym, including weight lifting exercises and the last training session took place three days prior to reaching the hospital. The training sessions were taken into account as a possible risk factor in her case [18].

### 2.5. Outcome and Follow Up

One month later she reported improvement in swallowing. However, her speech was still dysarthric. The soft palate palsy persisted along with paresis of the tongue. Due to low blood pressure values, the patient discontinued the antihypertensive drug. The dual antiplatelet therapy was maintained until the day of the next follow-up MRA examination. Two months after discharge home the patient started a neuro-logopaedic therapy based on the proprietary massage method of the oral cavity floor.

The follow-up MRA of the cerebral arteries presented a partial regression of the intramural hematoma of the left internal carotid artery with no significant narrowing of the arterial lumen. The pseudoaneurysm located just before entering the internal carotid artery canal was smaller than it was previously. There was no blood in the wall of the right vertebral artery (Figure 5).

The interventional radiologist maintained the lack of indications for endovascular therapy. The antiplatelet treatment was modified, leaving acetylsalicylic acid at 75 mg once daily. The blood pressure without medication remained within the normal range.

### 2.6. Neuro-Logopaedic Therapy

Before starting neuro-logopaedic therapy, the Frenchay Dysarthria Assessment scale revealed the following parameters: significant weakening of the cough reflex, a slight weakening of the swallowing and salivary reflexes, and an exorbitant gag reflex. Moreover, the patient had correct lip movements but experienced breathing disorders while speaking and suffered from soft palate dysfunction. Length, pitch, and strength of phonation were reduced to a significant extent during the above-mentioned examination. It also revealed tongue deviating to the left side with significant limitation of lateral and upward mobility. In addition, it presented atrophy of the tongue and moderate impairment of spontaneous speech.

As part of the therapy, we provided the patient with a special neurological massage of the sublingual muscles once a week. The stimulation started with the muscles located in the front part of the oral cavity. Then, after the gag reflex was reduced, the massage of the sublingual muscles located at the root of the tongue followed.

As a result, the patient experienced an improvement in the clarity of spontaneous speech and the reduction of breathing disturbances during speech. Subsequently, the range of tongue mobility increased. The massage provided periodically decreased hoarseness as well. Five months after the commencement of therapy, the patient’s speech significantly improved. Currently, all the assessed functions of the efficiency of the articulation organs and reflexes returned to normal conditions. Only a temporary slight decrease in the strength of voice still persists (Figure 6 and Figure 7).

## 3. Discussion

Internal carotid artery dissection is strongly associated with trauma [19,20,21], but it also occurs without any tangible connection with any trauma [14,22,23,24]. The haematoma dissecting the arterial wall penetrates between the middle and the inner membrane causing the inner membrane to bulge into the lumen of the artery. This is subsequently followed by a narrowing of its lumen and a predisposition to the formation of a parietal thrombus. This could eventually lead to acute cerebral ischemia in the area supplied by the ICA. In the course of formation, the intramural haematoma can also penetrate between the media and the adventitia, contributing to the formation of aneurysmal artery dilatation. The formation of pseudoaneurysm is associated with local symptoms of ICA dissection due to the compression of structures adjacent to the artery [25]. The dynamics of the described changes are usually related to the development of dissection symptoms over time [4], which was also the case in our patient.

The most commonly occurring local symptoms of internal carotid artery dissection and most strongly associated with the discussed pathology are headache, neck pain on the side of dissection (2/3 of patients), and Horner’s syndrome accompanied by ocular pain (25–58%). Approximately one-quarter of patients have a history of pulsating tinnitus [3,4,23].

A characteristic of internal carotid artery dissection is a headache located in the frontotemporal and facial area and less typically in the occipital area. Significant for the dissection is the fact that a huge number of patients (26%) complain of neck pain on the side of the damage. The pain, however, can mimic a migraine, which may delay obtaining a proper diagnosis, especially if the patient had previously experienced migraines [3,24,26,27,28].

Our patient experienced a headache the day before the tongue disorder appeared. It was a throbbing hemicrania that developed gradually and was accompanied by nausea and vomiting. However, there were no abnormalities on the neurological examination. The pain had features of migraine pain and it resolved after the application of treatment. Based on the clinical picture and the results of additional tests, no indications were leading to the suspicion of the symptomatic nature of pain [29]. Horner’s syndrome appeared two days after the onset of the headache and one day after the tongue swelling. It occurred among those already declared symptoms of damage to the lower (IX, X, XII) cranial nerves.

Symptoms of damage to the lower cranial nerves as a local symptom of dissection of the carotid artery are observed in 5–12% of patients with ICAD. In about 5.2% of cases, the damage to the lower cranial nerves (XII nerve, i.e., hypoglossal and additionally nerves IX, X, and XI in various combinations) is observed [3,30,31,32,33,34]. Internal carotid artery dissection is one of the possible causes of damage to the mentioned cranial nerves [35]. In the absence a neck injury or other risk factors for this type of pathology, it may be omitted in the initial differential diagnosis [23,25,36]. The problem may also be related to the correct diagnosis of symptoms resulting from damage to these nerves, which was the main factor resulting in the delay of the diagnosis in the case of the discussed patient.

On admission, the patient presented with asymmetric swelling of the tongue, which led to the differentiation from angioedema. Nevertheless, at that moment, she did not complain of any kind of headache. The patient’s report of difficulty in swallowing and hoarseness were part of this clinical picture recognition [37].

On the following day of observation, the tongue was symmetrical at rest. Left-sided tongue and soft palate paresis were evident. The patient also had paralysis of the left vocal fold and a lack of palatine and gag reflexes on that side. Such a picture did not raise any doubts as to the damage to the lower cranial nerves. It became clear that the previously observed left-sided tongue swelling was a symptom of damage to the hypoglossal nerve. Recognition of the hypoglossal nerve injury at this stage and considering the headache on the previous day would have likely allowed us to avoid delaying the proper diagnosis in our patient.

To gain a better understanding of this rare manifestation of ICAD, we reviewed five other cases in the literature to investigate the characteristics of the initial presentation and subsequent symptoms of this disease as well as the researchers’ conclusions. In Table 1 we presented a short summary of similar cases described by other researchers.

The rare phenomenon of isolated hypoglossal nerve palsy with tongue swelling can be the result of one of many pathologies that can affect the 12th nerve along its course from the brain stem to the tongue, in this case, a non-traumatic dissecting aneurysm of the internal carotid artery.

The analysis of the available literature shows that the case of our patient is not isolated. There have been a few reports of acute tongue swelling without paresis of the tongue as a manifestation of damage to the hypoglossal nerve in patients with dissection of the internal carotid artery. Fluid shift to the extracellular space or oedema after partial denervation due to compression of the hypoglossal nerve or its nutritional artery by aneurysmal dilatation of dissected internal carotid artery is thought to cause the tongue swelling [5,6,7,8,9,10].

The case of the patient presented in our research is another example of difficulties in making a prompt diagnosis in a patient with carotid artery dissection. The strength of this case report is mainly its educational value. Its limitation is that we may not generalize our findings.

## 4. Conclusions

Among the possible local symptoms of internal carotid artery dissection, proper attention should be paid to the possible atypical manifestation of damage to the hypoglossal nerve in the form of asymmetric swelling of the tongue. This is especially pertinent in cases where in an interview before tongue swelling, or along with it, the patient experiences a headache. The case of our patient may sensitize readers and thus facilitate the detection of similar or identical cases.

## Figures and Tables

**Figure 1 brainsci-13-00603-f001:**
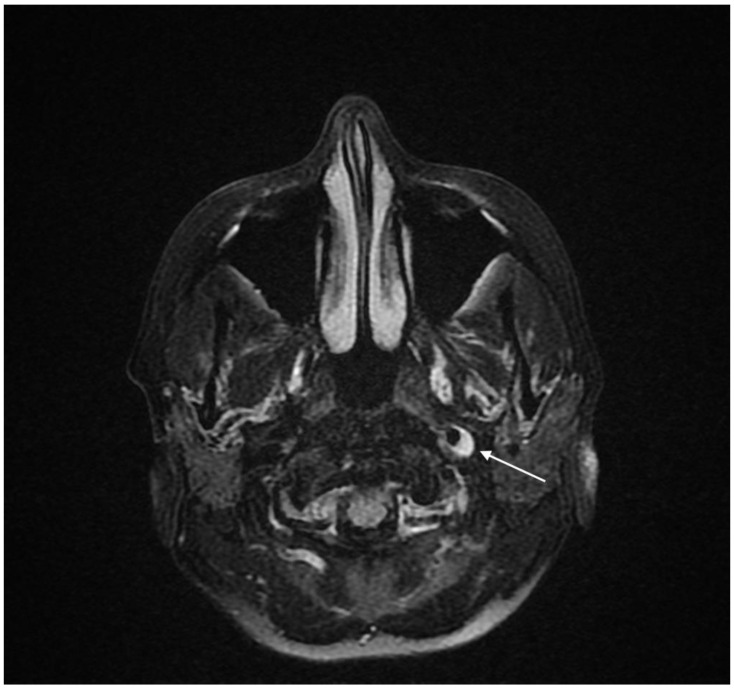
Baseline MRI of the brain. Axial fat-saturated FLAIR image clearly demonstrating an intramural thrombus in the left ICA (marked with an arrow).

**Figure 2 brainsci-13-00603-f002:**
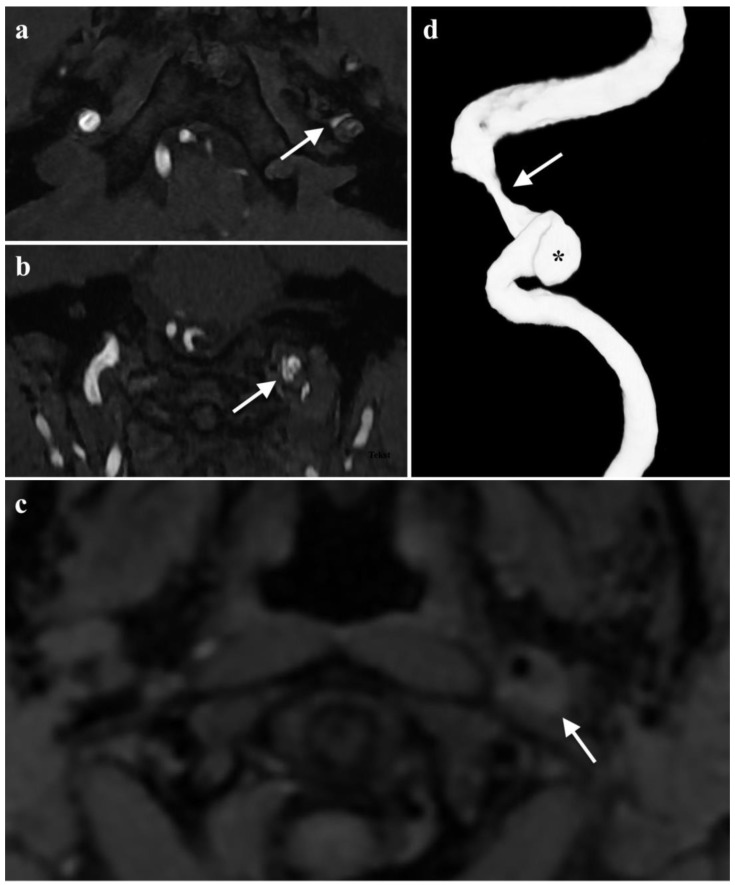
Baseline MRA of the head arteries: (**a**) TOF MRA image at the level of the carotid canal entrance showing the narrowing of the lumen of the left ICA (arrow); (**b**) TOF MRA image showing the false aneurysm of the left ICA (arrow); (**c**) prominent hyperintense crescent sign on the fat-saturated T1-weighted image caused by an eccentric intramural thrombus. The features are consistent with the diagnosis of left ICA dissection. (**d**) 3D VRT reconstruction demonstrates both the stenosis (arrow) and the false aneurysm (asterisk) of the left ICA.

**Figure 3 brainsci-13-00603-f003:**
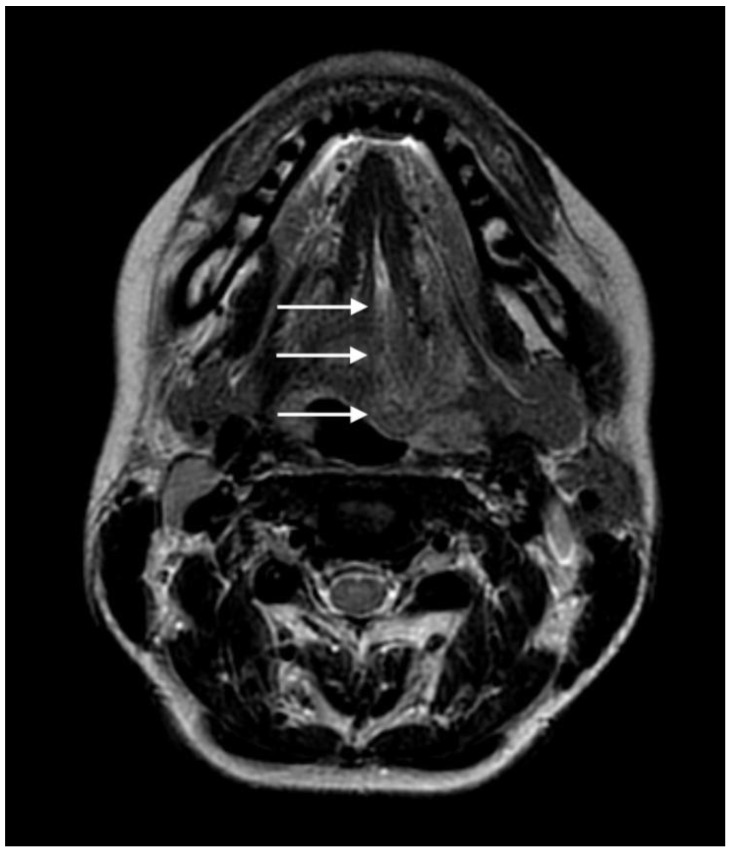
Axial T2-weighted image acquired during the baseline MRA of the head arteries. Axial T2-weighted image shows a swollen left hemitongue and increased T2 signal secondary to oedema (arrows).

**Figure 4 brainsci-13-00603-f004:**
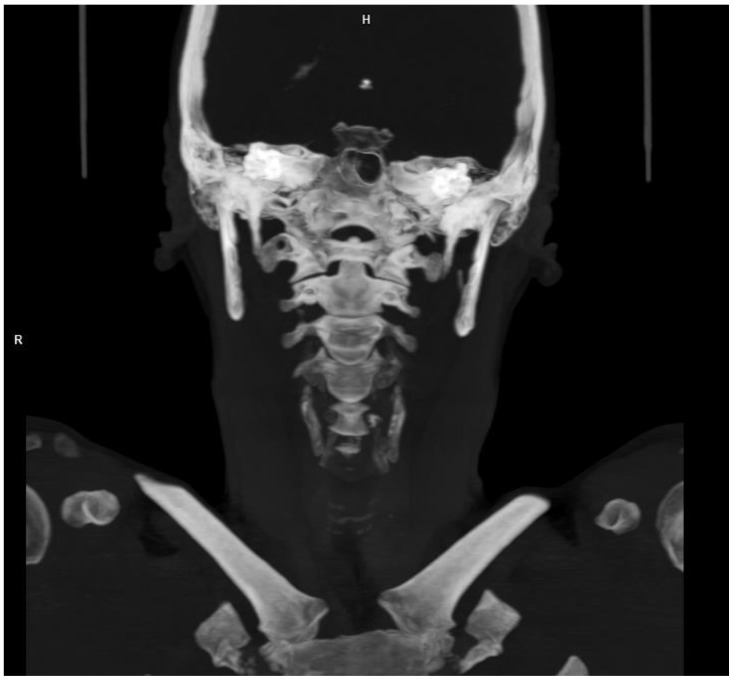
The styloid processes visualized on the CTA of the neck. The styloid processes length was in the normal range (15 mm bilaterally).

**Figure 5 brainsci-13-00603-f005:**
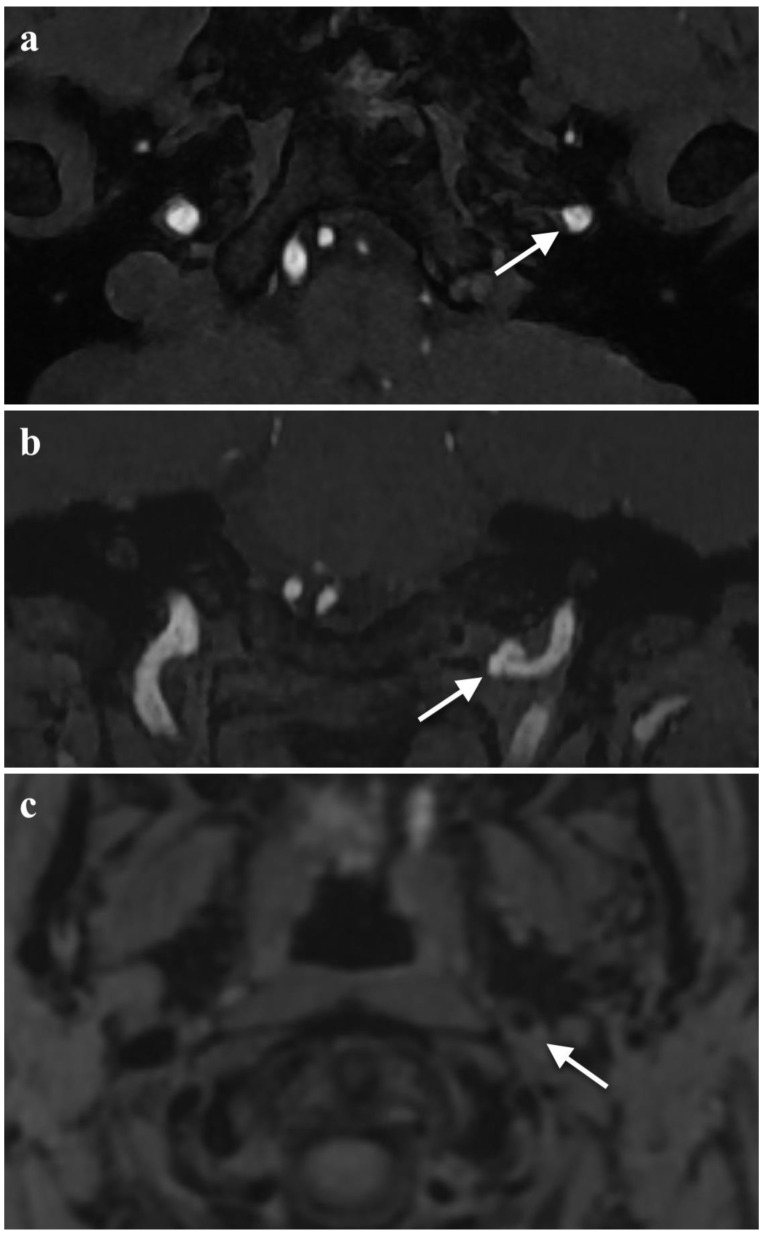
Follow-up MRA of the head arteries: (**a**) follow-up TOF MRA at the level of the carotid canal entrance showing the resolution of the left ICA stenosis (arrow); (**b**) follow-up TOF MRA demonstrating the reduction in the size of the left ICA false aneurysm (arrow); (**c**) fat saturated T1-weighted image showing small residual intramural thrombus in the left ICA (arrow).

**Figure 6 brainsci-13-00603-f006:**
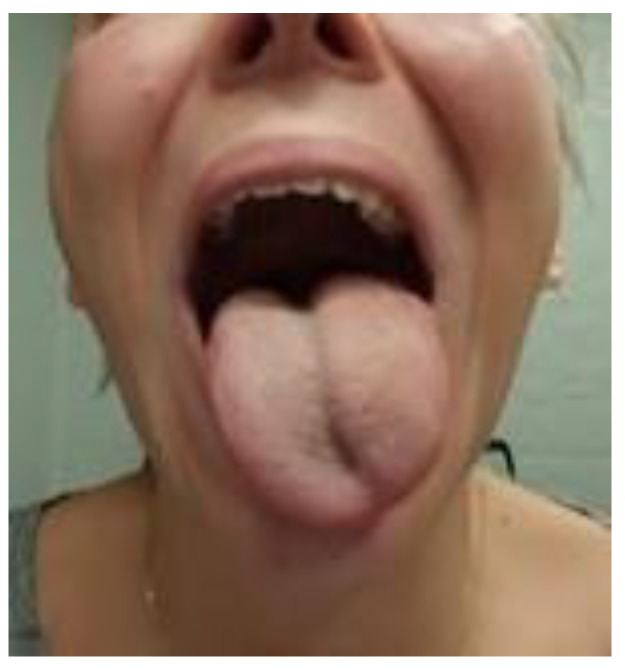
The patient’s photo before starting neuro-logopaedic therapy (January 2021). Visible left-sided pseudohypertrophy of the tongue and tongue deviation to the left while it is protruded.

**Figure 7 brainsci-13-00603-f007:**
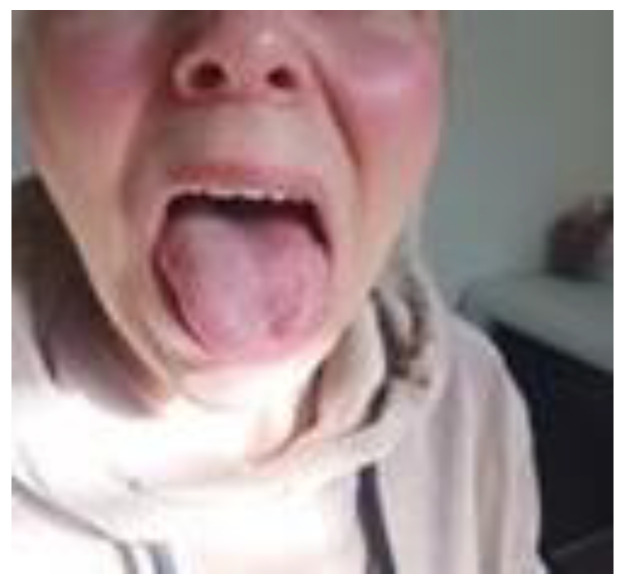
The patient’s photo three months later (April 2021).

**Table 1 brainsci-13-00603-t001:** Literature review regarding cases of carotid artery dissection with tongue swelling as one of the main symptoms.

Author and Year of Publication	Title	Type of the Study	Age of Described Subject	Gender of the Described Subject	Initial Symptoms of Internal Carotid Artery Dissection Present in the Described Subject				Subsequent Symptoms	Conclusions
					Tongue Swelling	Headache	Horner’s Syndrome	Cranial Nerves Injury Signs		
Ryan et al., 2015 [5]	Acute tongue swelling, the only initial manifestation of carotid artery dissection: a case report with differentiation of clinical picture	Case report	52 y. o.	Male	+	−	−	−	right-side tongue weakness (XII nerve palsy)	Hypoglossal palsy caused by ICAD should be considered in the differential diagnosis of patients who present with acute tongue swelling.
Siniscalchi et al., 2020 [6]	Carotid artery dissection induced acute tongue swelling in a cocaine user	Case report	42 y. o.	Male	+	−	−	−	left-sided tongue weakness and tongue atrophy on this side (XII cranial nerve palsy)	Arterial dissection was spontaneous and painless and this evidence combined with tongue swelling rather than the more expected weakness and atrophy led to the failure of recognition of the neurogenic nature of this swelling.
Engelbarts et al., 2017 [9]	Apparent swelling of the tongue	Clinical snapshot	51 y. o.	Male	+	+	−	−	left-sided tongue weakness with fasciculations (XII cranial nerve palsy), ipsilateral VII nerve palsy and vocal fold immobility on this side (X nerve palsy)	Cranial nerve involvement in carotid dissection is rare but led to the correct diagnosis in this case.
Kaushik et al., 2009 [10]	Spontaneous dissection of internal carotid artery masquerading as angioedema	Case report	44 y. o.	Male	+	−	−	−	slurred speech, left-sided tongue weakness (XII nerve palsy)	ICA dissection should be considered in the differential diagnosis of patients presenting with isolated hypoglossal nerve palsy and swelling of the tongue, although most cases present with local signs and ischemic stroke-like symptoms.
Shahab et al., 2001 [38]	Isolated hypoglossal nerve palsy due to internal carotid artery dissection	Case report	46 y. o.	Male	+	−	−	−	slurred speech, left-sided tongue weakness with fasciculations (XII cranial nerve)	The rare phenomenon of isolated hypoglossal nerve palsy with tongue swelling can be the result of one of many pathologies that can affect the 12th nerve along its course from the brain stem to the tongue, in this case, a non-traumatic dissecting aneurysm of the internal carotid artery.

+, presence of symptom; −, lack of symptom; y. o., years old; ICA, internal carotid artery; ICAD, internal carotid artery dissection.

## Data Availability

The datasets used and/or analysed during the current study are available from the corresponding author upon reasonable request.

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
