# Peer review of "Acute Tongue Swelling as a Still Unexpected Manifestation of Internal Carotid Artery Dissection: A Case Report"

_brainsci, 2023, doi:10.3390/brainsci13040603_

Round 1

Reviewer 1 Report

Authors present a case report on of a 36-year-old female with left internal carotid artery dissection (ICAD) with asymmetric left sided tongue swelling as atypical symptom. There are several cases reported in the literature of this unusual presentation. The patient did have a unilateral headache on the day previous to this symptom, which retrospectively should have been cleared by CT-Angiography where the ICAD would be presented, so unilateral tongue swelling was not the only symptom (she apparently had a Horner syndrome too), but given its rarity it deserves a special attention. I suggest to include a Table with all cases described in the literature and their pecularities. 

Reviewer 2 Report

Thank you for recommending me as a reviewer. The patient presented in this case study was a 36-year-old female without any comorbidities, not taking any medications on a regular basis and denying the use of stimulants. In this paper, she had no data of recent infections and head or neck traumas. Nevertheless, the day before the admission, the patient sought neurological care in our hospital due to a left-sided headache accompanied by emesis. At that time, no deviations in the neurological examination were found. Computed Tomography (CT) of the head revealed no pathologies. After receiving symptomatic treatment with improvement, she was discharged home. This case study is well written. If the authors complete minor revisions, the quality of the study will be further improved.

1. If the authors add implications for future research in the conclusion section, it can help readers understand.

2. The figures used in this study are adequate.
